# Mitigating Co Metal Particle Agglomeration and Enhancing ORR Catalytic Activity through Nitrogen-Enriched Porous Carbon Derived from Biomass

Yanling Wu [1], Qinggao Hou [1,*], Fangzhou Li [1], Yuanhua Sang [2], Mengyang Hao [1], Xi Tang [1], Fangyuan Qiu [1] and Haijun Zhang [3,*]

[1] Jinan Key Laboratory of New Energy & New Materials for Intelligent Transportation, School of Civil Engineering, Shandong Jiaotong University, Jinan 250357, China; wuyanling621@163.com (Y.W.)
[2] State Key Laboratory of Crystal Materials, Shandong University, Jinan 250100, China; sangyh@sdu.edu.cn
[3] Department of Vascular & Intervention, Tenth Peoples' Hospital of Tongji University, Shanghai 200072, China
[*] Correspondence: chqghou@163.com (Q.H.); zhanghaijun@tongji.edu.cn (H.Z.)

**Abstract:** Biomass-derived porous carbon has gained significant attention as a cost-effective and sustainable material in non-noble metal carbon-based electrocatalysts for the oxygen reduction reaction (ORR). However, during the preparation of transition metal catalysts based on biomass-derived porous carbon, the agglomeration of transition metal atoms often occurs, leading to a notable decline in catalytic activity. In this study, we present a straightforward synthetic approach for the preparation of nitrogen-enriched soybean-derived porous carbon (Co@SP-C-a) as an electrocatalyst for the ORR. To achieve this, we employed a two-step method. In the first step, a chemical activator (KCl) was utilized to enhance the porosity of the self-doped nitrogen biomass carbon material. In the second step, a constant pressure drop funnel technique was employed to uniformly disperse bimetal cobalt/zinc-based zeolitic imidazolium frameworks (ZIF-L and ZIF-67) containing different metal ions ($Zn^{2+}$ and $Co^{2+}$) into the activated biomass carbon material. Subsequent high-temperature calcination of the ZIF-L and ZIF-67@SP-C-a composite precursor yielded the Co@SP-C-a catalyst. The obtained catalyst exhibited remarkable ORR activity in an alkaline solution ($E_{onset}$ = 0.89 V, $E_{1/2}$ = 0.83 V, $J_L$ = −6.13 mA·cm$^{-2}$) and exceptional long-term stability. This study presents an effective strategy to prevent the agglomeration of metal nanoparticles when integrating them with biomass-based carbon materials, thus leading to enhanced catalytic performance.

**Keywords:** transition metal atoms; biomass-derived porous carbon; chemical activator; oxygen reduction reaction

## 1. Introduction

In recent decades, the increasing demands for energy, in parallel with the rapid economic growth, have raised significant concerns regarding the use of fossil fuels. The consequential energy and environmental challenges have prompted the establishment of ambitious goals, such as carbon peaking and carbon neutrality. Consequently, the development of sustainable and clean energy sources has become a crucial strategy for achieving these "dual carbon" objectives [1,2]. Notably, fuel cells and metal–air batteries have emerged as promising research areas due to their high energy density and environmentally friendly nature [3]. However, the oxygen reduction reaction (ORR), a critical process occurring in the electrode materials in fuel cells and metal–air batteries, presents inherent obstacles, including sluggish kinetics and high overpotential, leading to reduced overall efficiency [4,5]. Although Pt-based catalysts have been regarded as the most efficient ORR electrocatalysts, their high cost and poor stability pose significant challenges for widespread implementation [6]. Thus, there is an urgent need to develop inexpensive, efficient, and stable non-platinum-based ORR catalysts.

Biomass has emerged as a promising candidate material for various applications due to its abundance, low cost, large surface area, and three-dimensional porous structure. When subjected to high-temperature calcination, biomass materials exhibit excellent chemical stability, making them highly desirable for catalytic purposes [7,8]. However, pure biomass-based carbon materials alone are not optimal for ORR electrocatalysis due to their thicker carbon layers and inherent tendency to self-aggregate [9–11]. To improve the ORR activity of biomass-based carbon materials, the introduction of chemical activators and/or iron and/or cobalt transition metals into the carbon framework has been explored. Chemical activation, in particular, enables the creation of abundant edge sites and defect structures, thereby optimizing morphology and surface chemical states to improve ORR performance [12–15]. Commonly used chemical activators include KOH, $ZnCl_2$, and KCl/NaCl [16–19]. For instance, Zhu [19] employed KOH to activate carbonized spider protein to produce a new type of layered porous carbon nanosheets with a high nitrogen content. The material demonstrates a considerable specific surface area and elevated N content when subjected to an activation temperature of 800 °C. The insertion of AS-PC800 material into a symmetrical coin-shaped supercapacitor has demonstrated favorable performance in terms of rate capability and cycling stability. Furthermore, it can be observed that AS-PC800 demonstrates superior oxygen reduction capabilities, comparable to those of commercially available 20% Pt/C catalysts. This work presents a novel approach to investigate alternative biomass precursors to fabricate porous carbon materials with the potential to be used in energy conversion and storage. However, the toxicity and corrosiveness of KOH and $ZnCl_2$ pose limitations for large-scale production. As a result, the utilization of NaCl or KCl as molten salt activation technologies has gained significant attention. For example, Pan [20] described a technique for synthesizing N-doped porous carbon nanosheets (NCNS) using KCl-assisted pyrolysis. In the process of pyrolysis preparation, KCl crystals function as both templates and activators, facilitating the penetration of ZIF-8 derived carbon materials via capillary action. This, in turn, promotes the development of microporous and mesoporous carbon nanosheets. The manipulation of KCl quantity and pyrolysis temperature can significantly regulate the carbon material structure, nitrogen species, features, and degree of graphitization. The NCNS-10-900 catalyst, after optimization, demonstrates a greater half-wave potential and remarkable long-term stability in alkaline solutions. Additionally, its ORR performance is superior to that of Pt/C. The implementation of KCl-assisted pyrolysis MOF represents a viable approach to investigate effective and robust non-noble metal catalysts for energy conversion reactions. Alternatively, the impregnation of metal inorganic salts into biomass, followed by drying, grinding, and high-temperature carbonization, has also shown promise in improving catalytic performance for ORR. However, the extended durability of these catalysts remains a challenge due to the weak binding force between the metals and carbon materials.

Among the numerous methods, the utilization of metal–organic framework (MOF) materials for generating carbon-based ORR catalysts is regarded as a highly efficient strategy. This is due to the extensive range of compositions and structural variations offered by MOFs, enabling the creation of an ORR catalyst with optimal structural and compositional features [21–24]. Notably, zeolitic imidazolate framework (ZIF) materials have emerged as particularly promising precursors because they contain a sufficient number of nitrogen atoms within a highly porous structure [20,25–27]. However, as research deepens, researchers have discovered that single-component ZIFs have some drawbacks. For instance, ZIFs containing zinc result in porous carbon materials lacking mesoporous structures despite their high nitrogen content. Conversely, cobalt-containing ZIFs produce porous carbon with meso-pores and high conductivity but experience a decrease in nitrogen content during the graphitization process. These challenges significantly hinder the catalytic performance of ZIF-based materials in the ORR and, consequently, restrict their broader application in proton exchange membrane fuel cells. In addition, during high-temperature calcination, the easier spontaneous aggregation of metal atoms within single-component ZIFs leads to the formation of large metal particles, resulting in a reduced electrochemically active surface

area and active sites [28]. Recently, there has been a growing interest in Bi-metal–organic frameworks (MOFs), particularly zinc-based zeolitic imidazolium frameworks (Zn/Metal-ZIFs). These Zn/Metal-ZIFs have a large specific surface area due to the ability of zinc to evaporate and form a porous structure at a relatively low boiling point (907 °C). Moreover, the addition of zinc increases the proximity of other metals, acting as a "barrier" during the pyrolysis process and preventing the aggregation of metal nanoparticles.

Building upon the aforementioned insights, this study focuses on the design and preparation of well-dispersed cobalt-based biomass carbon materials with elevated nitrogen content, referred to as Co@SP-C-a, via a unique synthesis approach. Initially, the freeze-drying technique was employed, employing a chemical activator (KCl) to open pores within the self-doped nitrogen biomass carbon material. Subsequently, in order to obtain a carbon-based composite precursor wrapped in double ZIFs, the imidazole-modified KCl-activated biomass carbon material was uniformly impregnated with two distinct metal ion solutions using a constant pressure droplet funnel, resulting in the formation of a composite precursor known as ZIF-L&ZIF-67@SP-C-a. The Co@SP-C-a catalyst was then obtained through a secondary high-temperature calcination process at 900 °C, which was applied to the ZIF-LandZIF-67@SP-C-a composite precursor. Remarkably, the Co@SP-C-a catalyst exhibited exceptional ORR activity in an alkaline solution ($E_{onset}$ = 0.89 V, $E_{1/2}$ = 0.83 V, $J_L$ = −6.13 mA·cm$^{-2}$), along with outstanding long-term stability. This synthesis pathway, utilizing soybean flour as a self-doped template, KCl as a perforating agent, and an aqueous solution of bimetallic ZIF derivatives as a controlled-release evaporator, provides a valuable approach for obtaining highly dispersed cobalt-based materials for ORR applications.

## 2. Results and Discussion

To achieve uniform nanoparticle coverage on biological carbon materials, a novel device was designed in this study. The synthesis route of Co@SP-C-a, as illustrated in Figure 1, involved several steps. Initially, the commercial soybean powder (SP) was immersed in a potassium chloride solution, freeze-dried, and pyrolyzed to obtain activated carbon materials. Subsequently, a mixture of SP-a, Zn$^{2+}$, Co$^{2+}$, and 2-methylimidazole (Hmim) in deionized (DI) water was stirred, followed by drying and carbonization to produce the final product, Co@SP-C-a.

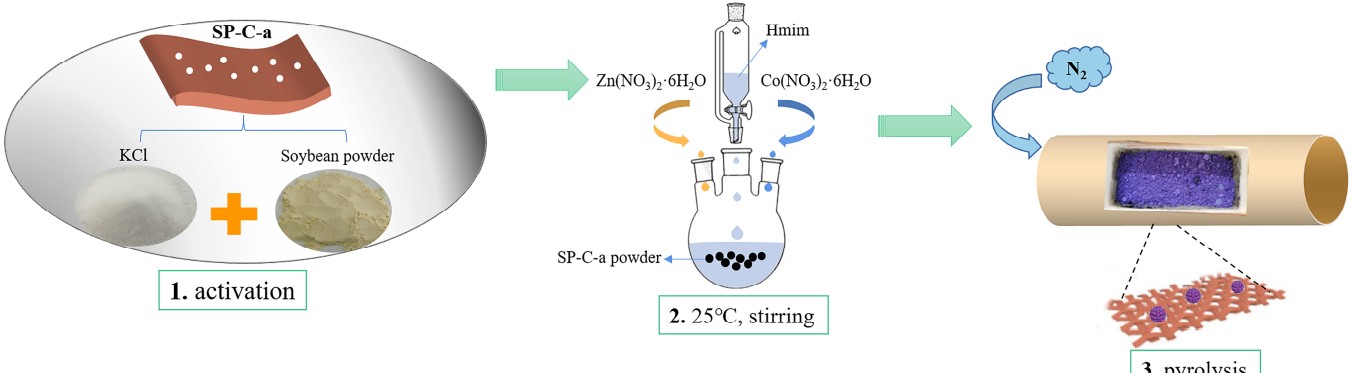

**Figure 1.** The synthetic scheme of Co@SP-C-a catalyst.

### 2.1. Characterization of Morphology and Structure

The morphological and structural component properties of Co@SP-C-a were examined through various techniques. Firstly, the X-ray diffraction (XRD) analysis (Figure 2a) confirmed that all three samples (SP-C, SP-C-a, and Co@SP-C-a) exhibited a wide (003) diffraction peak at around 26° and a distinct (101) diffraction peak at approximately 44°, indicating the presence of a graphite structure (Joint Committee on Powder Diffraction Standards (JCPDS) No. 26-1076). Then, the XRD pattern of SP-C-a revealed sharp peaks at 2θ = 29.4° and 51.4°, corresponding to the (301) and (130) facets of KNO$_3$ (JCPDS No. 41-0676), indicating the reaction between nitrate nitrogen in soybeans and potassium

ions during high-temperature activation to form $KNO_3$. The XRD pattern of the final Co@SP-C-a catalyst showed diffraction peaks attributed to Co (JCPDS No. 15-0806) and a graphitized carbon (GC) peak at around 26°. Notably, the GC (003) diffraction peak shifted to a higher angle after secondary high-temperature calcination of the ZIF-L&ZIF-67@SP-C-a precursor, indicating enhanced layer-to-layer interaction within the GC framework of Co@SP-C-a. This observation suggests that the presence of metal Co effectively promotes the degree of graphitization in the material [28].

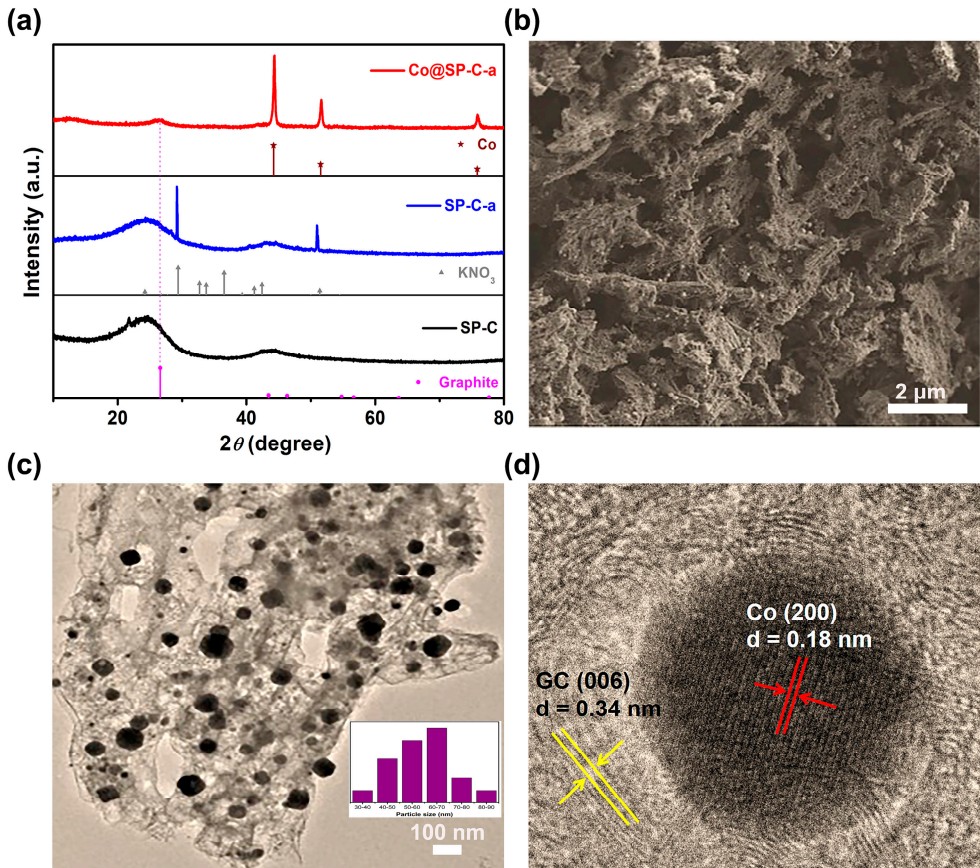

**Figure 2.** Representative XRD patterns (**a**); SEM (**b**), TEM (inset: particle size distribution of Co@SP-C-a) (**c**), and lattice fringe HR–TEM (**d**) images of Co@SP-C-a.

Next, the morphology of the as-prepared samples was analyzed using SEM. Firstly, the SEM image of SP-C revealed a surface with a concave and convex structure but without distinct pores (Figure S1). However, after KCl etching (SEM photographs of SP-C-a, Figure S2), an increase in the pore structure on the surface of SP-C-a was observed, indicating that KCl activation can indeed play a role in pore formation. Previous studies have reported that KCl not only protects the carbon structure from degradation at high temperatures, but also facilitates the formation of hierarchical porous structures during one-step carbonization, leading to a high specific surface area [20,29]. This hierarchical porosity could potentially enhance the exchange rate between the reaction products (such as oxygen) and electrolyte ions. Next, the SEM image and EDX mapping of the final Co@SP-C-a catalyst demonstrated a uniform distribution of Co elements (Figure 2b and Figure S3) through secondary high-temperature calcination while preserving the porous structure. The high-resolution TEM (HRTEM) image (Figure 2c) further provided additional evidence of the existence of a porous structure in Co@SP-C-a. According to the HRTEM image (Figure 2c), the surface of the final Co@SP-C-a catalyst was decorated with small inorganic quantum dots ranging from 30 to 90 nm in diameter, indicating the successful integration of the active inorganic species into the carbon material. Additionally, the high-resolution TEM (HRTEM) image

(Figure 2d) revealed lattice fringe spacings of 0.18 nm and 0.34 nm, matching the Co metal (200) and graphite carbon (006) planes, respectively.

The Raman spectrum of Co@SP-C-a (Figure 3a) exhibited two distinct peaks at around 1590 and 1340 cm$^{-1}$, in line with the G and D bands, respectively, suggesting the presence of ordered graphite carbon (G-C) and disordered carbon (D-C) in the sample [30]. The intensity ratio ($I_D/I_G$) of 0.963 suggests the incorporation of sp$^3$ hybridized G-C and D-C structures due to the co-doping of Co and N atoms. Furthermore, the nitrogen adsorption-desorption isotherms curves (Figure 3b) exhibited a dominant Type IV behavior with H$_3$ hysteresis, and an obvious hysteresis loop was observed when the P/P$_0$ exceeded 0.4. This confirmed the existence of mesoporous and microporous structures and a relatively high specific surface area of 331.59 m$^2$ g$^{-1}$ [31]. These results indicate that the Co@SP-C-a material possesses a combination of G-C and D-C, which can be attributed to the co-doping of Co and N atoms. The presence of mesoporous and microporous structures further contributes to its high specific surface area, which could be beneficial for various catalytic applications.

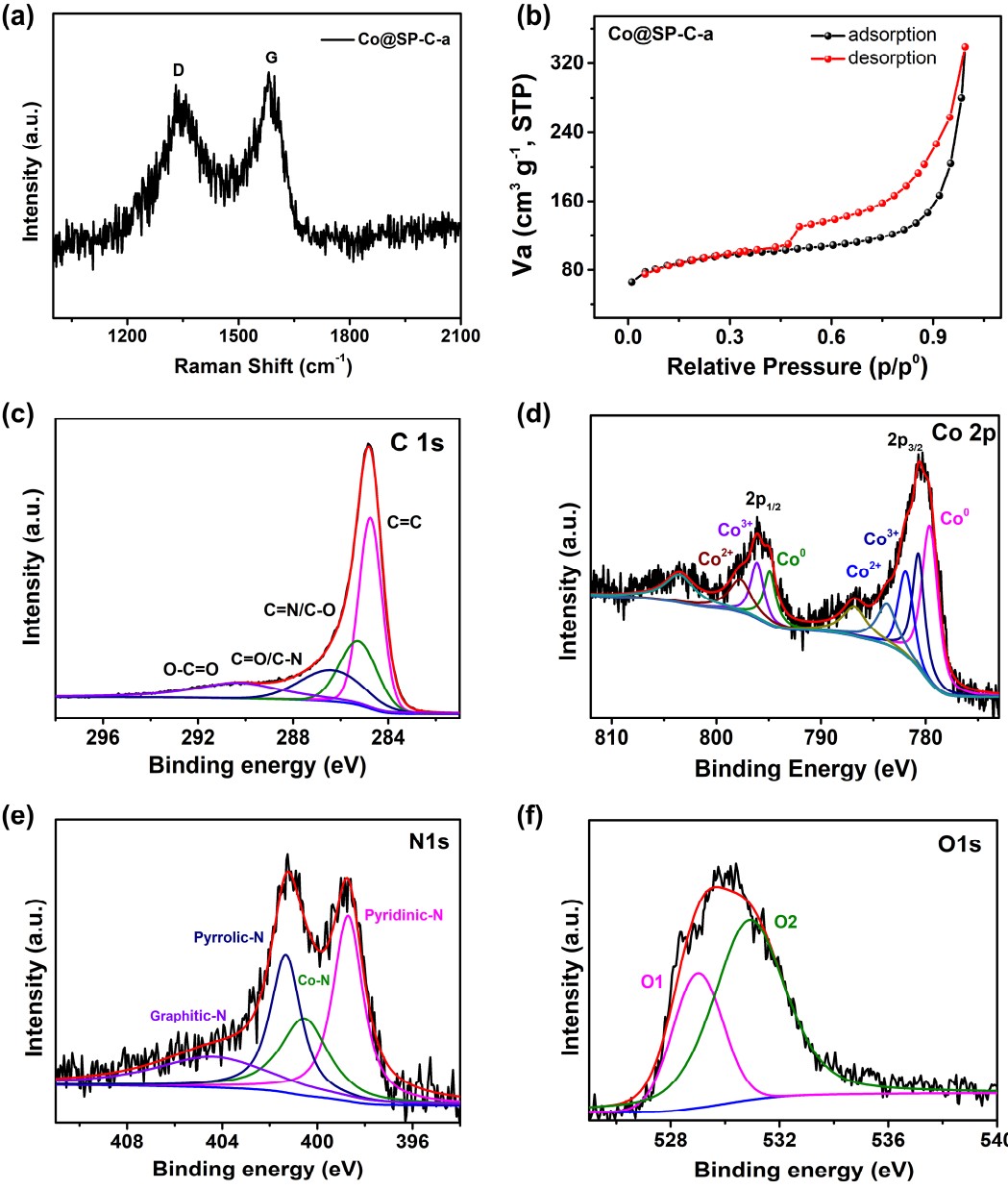

**Figure 3.** Raman spectra (**a**); the nitrogen adsorption/desorption isotherms (**b**); high-resolution XPS data for (**c**) C 1s, (**d**) Co 2p, (**e**) N 1s, (**f**) O 1s core levels of Co@SP-C-a.

Surface compositions and chemical valence states of Co@SP-C-a were characterized using XPS measurements (Figure 3c–f). The survey spectrum displayed distinctive peaks corresponding to C, N, O, and Co, with elemental contents of C (87.66%), N (6.43%), O (4.93%), and Co (0.98%) (Figure S4). The C 1s spectrum exhibited four distinct peaks with maxima at 284.8 (C=C), 285.3 (C=N/C–O), 286.4 eV (C=O/C–N), and 290.3 eV (O–C=O) (Figure 3c). The relatively high abundance of C=C double bonds suggests the presence of G-C, which increases electric conductivity and improves ORR activities [32]. The remaining three peaks observed in the C 1s spectrum are ascribed to defects and functional groups resulting from N atom doping in the carbon matrix [33,34].

In the Co 2p spectrum, two main peaks were observed around 780 eV and 796 eV (Figure 3d). These peaks can be further deconvoluted into nine branch peaks, representing $Co^0$ (779.6 eV, 794.9 eV), $Co^{3+}$ (780.7 eV, 796.1 eV), $Co^{2+}$ (781.9 and 797.7 eV), and three shakeup satellites (786.8 eV, 783.7 eV, and 803.5 eV). The formation of multivalent cobalt species can be attributed to the sensitivity of metallic cobalt to air oxidation [35]. The fitted N 1s spectra centered at 398.7, 400.5, 401.3, and 404.4 eV corresponded to pyridinic N, Co–N, pyrrolic N, and graphitic N, respectively (Figure 3e) [36,37]. The O 1s peaks at 529.0 (O1) and 530.9 eV (O2) in Co@SP-C-a may be attributed to the O-containing functional groups present on the C or Co surfaces (Figure 3f) [38]. These XPS results provide valuable insights into the surface chemistry of Co@SP-C-a, highlighting the presence of graphitic carbon, nitrogen heteroatom doping, and the different oxidation states of cobalt. These surface characteristics are expected to play a crucial role in determining the catalytic performance and electrochemical properties of the material.

### 2.2. Electrochemical Properties

The electrocatalytic performance of the catalysts (SP-C, SP-C-a, and Co@SP-C-a) for ORR was evaluated using CV in 0.1 M potassium hydroxide saturated with oxygen and nitrogen at a scan rate of 50 mV/s (Figure 4a and Table S1). Among the catalysts tested, Co@SP-C-a exhibited a significantly more positive peak potential and the highest ORR activity. This observation suggests that the active ingredients in Co@SP-C-a play a crucial role in enhancing the catalytic activity for ORR [39]. These findings highlight the potential of Co@SP-C-a as a promising catalyst for ORR and warrant further investigation to understand the underlying mechanisms contributing to its superior performance.

The ORR activities of the catalysts were evaluated using LSV polarization curves at a scan rate of 10 mV/s (Figures 4b, S5 and S6 and Table S1). Notably, Co@SP-C-a exhibited superior ORR performance compared to the other samples. The ORR activity order was as follows: Co@SP-C-a ($E_{onset}$ = 0.89 V, $E_{1/2}$ = 0.83 V, $J_L$ = −6.13 mA·cm$^{-2}$) > SP-C-a ($E_{onset}$ = 0.86 V, $E_{1/2}$ = 0.78 V, $J_L$ = −4.45 mA·cm$^{-2}$) > SP-C ($E_{onset}$ = 0.76 V, $E_{1/2}$ = 0.57 V, $J_L$ = −2.58 mA·cm$^{-2}$). LSV profiles in the rotating disk electrode (RDE) configuration were used to compare the catalytic activity of Co@SP-C-a produced by pyrolysis at 800 °C and 1000 °C (designated Co@SP-C-a-800 and Co@SP-C-a-1000, respectively), as illustrated in Figures S7 and S8 and Table S1. The catalytic activity of ORR was found to be lower in comparison to Co@SP-C-a when subjected to pyrolysis temperatures below or above 900 °C. The observed outcome can be attributed to the generation of ZnO under inadequate activation conditions, specifically at a temperature of 800 °C, which resulted in incomplete exposure of the active site. At a temperature of 1000 °C, the catalyst experienced a high activation energy, causing it to undergo skeleton shedding. This led to structural collapse and hindered the transfer of mass into the solution. These results suggest that further treatment of the SP-C materials enhances the catalysts' properties. Remarkably, Co@SP-C-a carbonized at 900 °C exhibited high onset potential, high half-wave potential, and a limiting current density comparable to Pt/C catalyst.

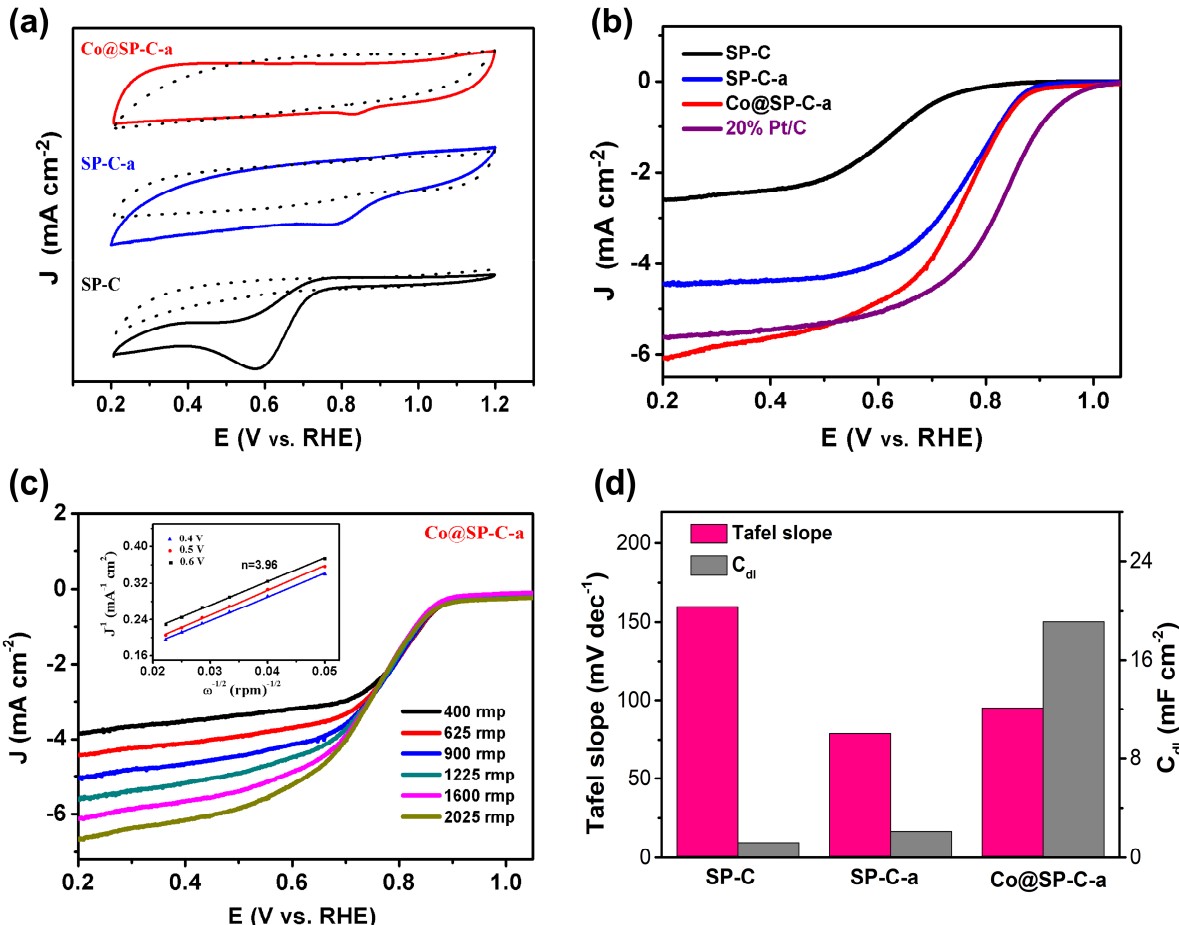

**Figure 4.** (**a**) CV curves obtained for SP-C, SP-C-a, and Co@SP-C-a at 50 mV s$^{-1}$ in 0.1 M potassium hydroxide saturated with O$_2$ (solid lines) and N$_2$ (dotted lines). (**b**) ORR LSV curves at 10 mV s$^{-1}$ under 1600 rpm recorded for the three SP-C-based catalysts and a 20 wt.% Pt/C catalyst in the same electrolyte saturated with oxygen. of (**c**) LSV curves of Co@SP-C-a at 10 mV s$^{-1}$ under various rotation speeds, with the inset showing K-L plots of Co@SP-C-a at different potentials. (**d**) Tafel slope and electrochemical double-layer capacitance (C$_{dl}$) comparisons of SP-C, SP-C-a, and Co@SP-C-a.

To obtain detailed information on the ORR mechanism of Co@SP-C-a, RDE measurements of Co@SP-C-a were conducted at various rotation rates (400–2500 rpm), showing a high current density at high rotation rates (Figure 4c). The number of electrons transferred during the catalytic reaction was calculated based on the Koutecky-Levich (K-L) equation [40]. The linearity and parallel values of the K-L plots in the potential range from 0.4 to 0.6 V suggest that the Co@SP-C-a catalyst displayed first-order reaction kinetics in response to the concentration of dissolved oxygen. The number of transferred electrons (n) of Co@SP-C-a was determined to be 3.96 (inset of Figure 4c). This value was close to the n values for Pt/C (4.0) and significantly higher than those of SP-C (1.96) and SP-C-a (3.29), as shown in Table S1. These findings strongly suggest that Co@SP-C-a follows a four-electron O$_2$ reduction mechanism, similar to that of Pt/C [41]. Meanwhile, compared to SP-C (160 mV dec$^{-1}$), the Tafel slope value of Co@SP-C-a (95 mV dec$^{-1}$) was similar to Pt/C (97 mV dec$^{-1}$), indicating an effective electron transfer, but it was greater than the Tafel slope value of SP-C-a (75 mV dec$^{-1}$). Furthermore, the electrical double-layer capacitance (C$_{dl}$), acquired through the CV curves at various scan rates in the potential range from 1.0.5 to 1.15 V, was used to estimate the electrochemical surface area (ECSA) [42,43]. Compared with SP-C (C$_{dl}$ = 1.22 mF cm$^{-2}$) and SP-C-a (C$_{dl}$ = 2.10 mF cm$^{-2}$), a higher C$_{dl}$ value was acquired for the Co@SP-C-a catalyst (19.12 mF cm$^{-2}$), indicating there were more accessible surface reactive sites, which are conducive to the enhanced activity during the ORR process

(Figures [4]d, S9 and S10). This indicates the favorable catalytic activity and high efficiency of Co@SP-C-a, further supporting its potential as a promising alternative to Pt/C in ORR applications. The ability of Co@SP-C-a to facilitate the four-electron reduction process is crucial for its practical utilization in energy conversion devices and underscores the significance of incorporating cobalt into the SP-C framework.

The long-term stability of ORR catalysts is a crucial factor in evaluating their efficacy. To assess the stability of Co@SP-C-a and Pt/C, we conducted time-amperometric tests at 1600 rpm and 0.8 V in $O_2$-saturated 0.1 M potassium hydroxide. Remarkably, after repeated measurements for 10,000 s, Co@SP-C-a retained 86% of the initial current density, while Pt/C only retained 73% of its initial current density (Figure S11). This finding highlights the higher stability of the Co@SP-C-a catalyst compared to Pt/C. The enhanced stability of Co@SP-C-a can be attributed to the presence of nitrogen-doped carbon, which provides stable anchoring sites for the active component of cobalt, thereby improving the dispersibility of the carbon carrier and active components [44–46]. This nitrogen-doping strategy contributes to the superior stability of Co@SP-C-a. Furthermore, it is worth noting that the ORR performance of Co@SP-C-a surpasses the majority of electrocatalysts reported in the literature, emphasizing its potential as an outstanding candidate for ORR applications (Table S2).

## 3. Materials and Methods

### 3.1. Materials

Zinc nitrate hexahydrate ($Zn(NO_3)_2 \cdot 6H_2O$), cobalt(II) nitrate hexahydrate ($Co(NO_3)_2 \cdot 6H_2O$), potassium hydroxide (KCl), and 2-methylimidazole (Hmim) were purchased from Aladdin Industrial Corporation. Ethanol was purchased from Kelong Chemicals. Pt/C (20 wt%) electrocatalyst and Nafion solution (5 wt%, D520, HiSPEC®R3000) were purchased from Alfa Aesar chemical Co., Ltd (Shanghai, China). The soybean powder utilized in this study was purchased from Jiaxiang Yongsheng Food Co., Ltd (Jining, China). and had not undergone any processing.

### 3.2. Synthesis of SP-C

To examine the properties of the biomass itself and establish a comparative reference, soybean powder (referred to as SP) underwent high-temperature calcination. A designated amount of pure soybean powder (100 mg) was introduced into a tube furnace and subjected to heating at 900 °C for 2 h under a flowing $N_2$ atmosphere with a flow rate of 5 °C/min. After cooling, the resulting powder was stirred in 6 M HCl at 80 °C for 4 h to eliminate impurities. Subsequently, it was rinsed with DI $H_2O$ until the pH reached approximately 7.0 and was then vacuum-dried, resulting in the formation of SP-C.

### 3.3. Synthesis of SP-C-a

To synthesize SP-C-a, a mixture of soybean powder (1.0 g) and KCl (1.0 g) was prepared in 20 mL of DI water and stirred for 30 min. The mixture was then subjected to freeze-drying to remove moisture. Subsequently, the product was transferred into a ceramic boat and placed in a tube furnace. The furnace was heated to 900 °C at a ramping rate of 5 °C/min, and the mixture was maintained at this temperature for 2 h under an $N_2$ atmosphere. This process yielded SP-C-a, with a mass ratio of 1:1 (where "a" indicates biomass activated by KCl). The obtained product was immersed in 6 M HCl and stirred for 4 h at 80 °C. It was then washed with DI water until the pH reached approximately 7.0 and was subsequently vacuum-dried overnight.

### 3.4. Synthesis of Co@SP-C-a-800, Co@SP-C-a, Co@SP-C-a-1000

The synthesis procedure for the ZIF-Land ZIF-67 (containing $Zn^{2+}$ and $Co^{2+}$ metal ions) aqueous solution followed a typical protocol [47,48]. Initially, zinc nitrate (0.5 mmol), cobalt nitrate (0.5 mmol), and SP-C-a powder (150 mg) were dissolved in 20 mL of DI water at a 1:1 mass ratio. The suspension was stirred continuously at r.t. for 30 min. Subsequently,

Hmim (8 mmol, 20 mL) was added to the carbon black solution using a constant pressure drop funnel. The resulting mixture, denoted as ZIF-Land ZIF-67@SP-C-a, exhibited a purple color. The collected compound was then subjected to a high-temperature treatment at 900 °C with a ramping rate of 5 °C/min for 2 h under a nitrogen atmosphere. The product was immersed in 6 M HCl and stirred at 80 °C for 4 h, followed by washing with DI water until the neutral pH value was reached. Finally, the product was vacuum-dried overnight to obtain Co@SP-C-a. Similarly, change the temperature of SP-C-a (800 °C and 1000 °C, respectively) to obtain Co@SP-C-a-800 and Co@SP-C-a-1000, respectively.

### 3.5. Material Characterization

The samples were subjected to X-ray diffraction (XRD, Bruker, Karlsruhe, Germany) analysis in a laboratory setting using a Rigaku Ultima IV X-ray diffractometer (Rigaku, Tokyo, Japan) equipped with a Cu Kα source (40 kV, 40 mA). The morphology and structure of the samples were examined using a field-emission scanning electron microscope (SEM, Quant 250FEG, Hamburg, Germany) coupled with an energy-dispersive X-ray (EDX) detector and a high-resolution transmission electron microscope with an acceleration voltage of 200 kV (TEM, JEM-2100F, JEOL Ltd., Tokyo, Japan). The Brunauer–Emmett–Teller (BET) surface area and pore size distribution (PSD) were measured using the Micromeritics Belsorp-max analyzer (Unterschleißheim, Germany). X-ray photoelectron spectroscopy (XPS) experiments were performed on Kratos Axis Ultra equipment (Manchester, UK) employing monochromatic Al Kα radiation. Raman scattering spectra were collected using a laser Raman microscope system (Nanophoton RAMANtouch, Osaka, Japan) with an excitation wavelength of 532 nm.

### 3.6. Electrochemical Measurements

The electrochemical measurements were conducted utilizing a standard three-electrode configuration on a Gamry (RDE710, Warminster, PA, USA) electrochemical workstation. The reference and counter electrodes employed were the Ag/AgCl (KCl-saturated) electrode and a carbon rod, respectively. To assure the repeatability of the experiment, the working electrode for each of the four catalysts was prepared using identical conditions. The procedure for the preparation of a working electrode involved dispersing 5 mg of catalyst powder in a mixture of 0.8 mL of ethyl alcohol and 40 μL of Nafion solution (5 wt%, Sigma-Aldrich, Shanghai, China) through sonication to achieve a uniform suspension. Following that, 10 μL (mg·cm$^{-2}$) of catalyst ink was dropped onto the surface of the glassy carbon electrode. For the ORR, all measurements were carried out in 0.1 M KOH solution. Cyclic voltammetry (CV) measurements were obtained in $N_2$-saturated or $O_2$-saturated solution at a scan rate of 50 mV s$^{-1}$. Linear sweep voltammetry (LSV) was performed using a rotating disk electrode (RDE) at a scan rate of 10 mV s$^{-1}$ with varying rotation speeds (400, 625, 900, 1225, 1600, and 2025 rpm) and 90% iR compensation. The Tafel slopes were determined from the linear fitting of the polarization curves according to the Tafel equation: η = blogj + a, where j is the current density and b denotes the Tafel slope. The stability of ORR in an $O_2$-saturated 0.1 M KOH solution was evaluated using a current versus time (i–t) measurement at 1600 rpm. The ORR performance of the as-prepared catalysts was compared with that of a commercially available, state-of-the-art Pt/C (20 wt%) electrocatalyst. The electron transfer number and kinetic current density ($J_K$) of the ORR reaction were calculated using the Koutecky-Levich equation [49,50]:

$$1/J = 1/J_L + 1/J_K = 1/B\omega^{1/2} + 1/J_K \tag{1}$$

$$B = 0.62nFC_0(D_0)^{2/3}\nu^{-1\backslash6} \tag{2}$$

$$J_K = nFkC_0 \tag{3}$$

In these equations, J represents the current density in the LSV curve, while $J_L$ and $J_K$ represent the diffusion-limiting current density and the kinetic-limiting current density,

respectively. B represents the reciprocal of the slope from the K-L curve, and $\omega$ represents the electrode rotation rate. $C_0$ ($1.2 \times 10^{-6}$ mol cm$^{-3}$) and $D_0$ ($1.9 \times 10^{-5}$ cm$^2$ s$^{-1}$) represent the bulk concentration and diffusion coefficient of oxygen in the 0.1 M KOH solution, respectively. F denotes the Faraday constant (96,485 C mol$^{-1}$), $\nu$ represents the dynamic viscosity, and k represents the rate constant of electron transfer.

## 4. Conclusions

We have successfully developed a straightforward synthetic approach to fabricate a dispersed cobalt-based nitrogen-enriched porous carbon catalyst for ORR. The utilization of freeze-drying technology ensures the uniform drilling of holes on the biomass carbon template by chemical activators, while the MOF materials self-grow on the activated carbon template through dropwise titration. By subjecting the resulting materials to secondary high-temperature carbonization, we obtained a high-performance ORR catalyst, Co@SP-C-a. This catalyst exhibits remarkable catalytic activity in alkaline conditions ($E_{onset}$ = 0.89 V, $E_{1/2}$ = 0.83 V, $J_L$ = −6.13 mA·cm$^{-2}$), along with excellent long-term stability. Importantly, our work presents innovative strategies for effectively preventing the agglomeration of metal nanoparticles.

**Supplementary Materials:** The following supporting information can be downloaded at: https://www.mdpi.com/article/10.3390/catal13071118/s1, Figure S1: The SEM image of SP-C; Figure S2: The SEM image of SP-C-a; Figure S3: SEM image of Co@SP-C-a used in the EDS mapping area revealing the elemental distribution of Co, C, N, and O; Figure S4: XPS survey spectra of Co@SP-C-a; Figure S5: LSV curves of SP-C at various rotating speeds, respectively. (Inset: K-L plots of SP-C at various potentials.); Figure S6: LSV curves of SP-C-a at various rotating speeds, respectively. (Inset: K-L plots of SP-C-a at various potentials.); Figure S7: CV and LSV curves of SP-C-a-800 at various rotating speeds, respectively. (Inset: K-L plots of SP-C-a-800 at various potentials.); Figure S8: CV and LSV curves of SP-C-a-1000 at various rotating speeds, respectively. (Inset: K-L plots of SP-C-a-1000 at various potentials.); Figure S9: Tafel plots of SP-C, SP-C-a, Co@SP-C-a, and 20% Pt/C were obtained via LSV data; Figure S10: Cyclic voltammograms (CV) at various scan rates; Figure S11: Amperometric i–t curves of Co@SP-C-a and 20 wt% Pt/C in O$_2$-saturated 0.1 M KOH solution with the rotation speed of 1600 rpm; Table S1: The ORR performance of the SP-C, SP-C-a, Co@SP-C-a-800, Co@SP-C-a-1000, and Co@SP-C-a, in alkaline media at 1600 rpm, respectively; Table S2: Comparison of the ORR performance for Co@SP-C-a catalysts at 1600 rpm in 0.1 M KOH. References [51–61] are cited in the supplementary materials.

**Author Contributions:** Writing—original draft and writing—review and editing, Y.W.; investigation, F.L. and F.Q.; data curation and supervision, X.T. and M.H.; funding acquisition, H.Z.; resources, Q.H. and Y.S. All authors have read and agreed to the published version of the manuscript.

**Funding:** This work is financially supported by the Innovation Training Foundation for College Student of Shandong Jiaotong University and the Natural Science Foundation of Shandong Province (ZR2021MB027).

**Data Availability Statement:** Data sharing is not applicable to this article.

**Conflicts of Interest:** There are no conflict to declare.

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
