# Peer review of "Mitigating Co Metal Particle Agglomeration and Enhancing ORR Catalytic Activity through Nitrogen-Enriched Porous Carbon Derived from Biomass"

_catalysts, doi:10.3390/catal13071118_

Round 1

Reviewer 1 Report

This study is to introduce biomass-derived porous carbon substrates for the oxygen reduction reaction, showing a cost-effective and sustainable approach potentially in fuel cell applications. Even if its results are not outperformed compared to the cost-effective method, it should be accepted by considering the following comments.

1. Sections 2 and 3 seem to be reversed. “3. Materials and Methods” should be changed to “2. Materials and Methods” and Figure 1 remains in Section 2 (Materials and Methods). KOH in Figure 1 should be changed to KCl.

2. More detailed explanation of experimental methods should be done. For example, how are the soybean powders prepared before mixing with the KCl solution? Please specify the manufacture of commercial Pt/C. What are the reasons for conducting “a secondary high-temperature calcination process” (Line 105)? What is Hmim? It is suggested to define all abbreviations, such as Hmin to 2-methylimidazole (Hmim).  

3. In Figure 3, changing the intensities to arbitrary units (a.u.) should be considered as the values may not be meaningful. The K-L method (Koutechy–Levich) should be described with its equation and a reference in the main text before discussing the ORR activity in Figure 4c. 

Minor grammatical corrections may make the quality of the manuscript improved.

Author Response

From:  Prof. Dr. Haijun Zhang

Department of Vascular & Intervention, Tenth Peoples’ Hospital of Tongji University Shanghai 200072

  1. R. China

E-mail:zhanghaijun@tongji.edu.cn.

Dr. Yanling Wu

School of Transportation and Civil Engineering, Shandong Jiaotong University

Ji'nan 250357

  1. R. China

E-mail: wuyanling621@163.com.

July 8, 2023

Dear Laszlo Szabolcs Kohr,

Many thanks for your kind help in dealing with our following paper with the manuscript ID: catalysts-2475268 entitled “Mitigating Co Metal Particle Agglomeration and Enhancing ORR Catalytic Activity through Nitrogen-Enriched Porous Carbon Derived from Biomass”. Please find the revised manuscript together with the supplementary data attached. According to the editor and the referees’ instructions, we have made corresponding revisions highlighted in blue over the whole manuscript to clarify the points of the referees. Also, please find our responses to the editor and the referees’ comments point by point below.

We wish this revised version of manuscript could meet the standards of the editor and the referees and your journal, and thus can be published in Catalysts.

Thanks again for your kind help and cooperation.

Kindest regards and wishes,

Yanling Wu and Haijun Zhang

Response to reviewers’ reports:

Response: Many thanks for the comments and suggestions of this referee, which have been strictly followed during the revision process of the present manuscript.

Reviewer 1:

This study is to introduce biomass-derived porous carbon substrates for the oxygen reduction reaction, showing a cost-effective and sustainable approach potentially in fuel cell applications. Even if its results are not outperformed compared to the cost-effective method, it should be accepted by considering the following comments.

  1. Sections 2 and 3 seem to be reversed. “3. Materials and Methods” should be changed to “2. Materials and Methods” and Figure 1 remains in Section 2 (Materials and Methods). KOH in Figure 1 should be changed to KCl.

Response: Many thanks for your careful review and comments. The above questions have been corrected in the revised version of the manuscript.

  1. More detailed explanation of experimental methods should be done. For example, how are the soybean powders prepared before mixing with the KCl solution? Please specify the manufacture of commercial Pt/C. What are the reasons for conducting “a secondary high-temperature calcination process” (Line 105)? What is Hmim? It is suggested to define all abbreviations, such as Hmin to 2-methylimidazole (Hmim).

Response: Thanks a lot for the comments. The issues mentioned above have been modified as required in the revised version of the manuscript, please see “Materials and Methods” section (page 3-4, line 131–144.). In addition, the reason for the secondary high-temperature calcination of the ZIF-L&ZIF-67@SP-C-a composite precursor is to further improve conductivity and generate more defect sites.

  1. In Figure 3, changing the intensities to arbitrary units (a.u.) should be considered as the values may not be meaningful. The K-L method (Koutechy–Levich) should be described with its equation and a reference in the main text before discussing the ORR activity in Figure 4c.

Response: Thank you very much for your kind reminding. The intensity values of the Raman spectrum and the XPS spectra have been removed in Figure 3. The K-L method (Koutechy–Levich) has been described with its equation (page 5, line 207–219.) and references (ref. 30, 31).

Reviewer 2 Report

The manuscript titled “Mitigating Co Metal Particle Agglomeration and Enhancing ORR Catalytic Activity through Nitrogen-Enriched Porous Carbon Derived from Biomass” presents a synthetic approach for the preparation of nitrogen-enriched soybean-derived porous carbon as an electrocatalyst for the ORR. This work contributes significantly to the field and offers promising and relevant findings within the scope of the Journal. However, there are several issues that need to be addressed before considering publication in Catalysts.

1. Remove the phrase 'Type of the Paper' from the beginning of the manuscript.

2. The authors should enhance the introduction by highlighting the significance of ZIG as precursors for synthesizing heteroatom-doped porous materials in this particular field.

3. In my opinion, Figure 1 should be included under the Methodology section.

4. Could you please provide the size of Co@Sp-C-a? Additionally, it would be valuable to discuss the conclusions that can be drawn from the XRD patterns, SEM, and TEM images.

5. What was the rationale behind selecting a scan rate of 50 mV/s for analyzing the electrochemical properties, as opposed to 10 or 20 mV/s? Similarly, what motivated the choice of 0.1 M KOH electrolyte? These aspects warrant further clarification in the manuscript.

6. Could you please provide more information regarding the current density target of the process? Elaborating on this important aspect would enhance the understanding of the research findings.

7. It is recommended to enhance the quality of the figures presented in the manuscript to improve their clarity and visual appeal.

8. The revised version of the manuscript should include a dedicated materials subsection to provide detailed information about the materials used in the study.

9. Could you please provide an explanation for the observed improvement in results compared to Pt-based catalysts? It would be highly informative to understand the underlying phenomena driving this process, including aspects such as the Tafel slope, mass activity, and other relevant parameters.

10. It is recommended to include additional references in the revised version of the manuscript to further support the presented findings.

11. Could you please provide information about the composition of the electrocatalyst?

Author Response

From:  Prof. Dr. Haijun Zhang

Department of Vascular & Intervention, Tenth Peoples’ Hospital of Tongji University Shanghai 200072

  1. R. China

E-mail:zhanghaijun@tongji.edu.cn.

Dr. Yanling Wu

School of Transportation and Civil Engineering, Shandong Jiaotong University

Ji'nan 250357

  1. R. China

E-mail: wuyanling621@163.com.

July 8, 2023

Dear Laszlo Szabolcs Kohr,

Many thanks for your kind help in dealing with our following paper with the manuscript ID: catalysts-2475268 entitled “Mitigating Co Metal Particle Agglomeration and Enhancing ORR Catalytic Activity through Nitrogen-Enriched Porous Carbon Derived from Biomass”. Please find the revised manuscript together with the supplementary data attached. According to the editor and the referees’ instructions, we have made corresponding revisions highlighted in blue over the whole manuscript to clarify the points of the referees. Also, please find our responses to the editor and the referees’ comments point by point below.

We wish this revised version of manuscript could meet the standards of the editor and the referees and your journal, and thus can be published in Catalysts.

Thanks again for your kind help and cooperation.

Kindest regards and wishes,

Yanling Wu and Haijun Zhang

Response to reviewers’ reports:

Response: Many thanks for the comments and suggestions of this referee, which have been strictly followed during the revision process of the present manuscript.

Reviewer 2:

The manuscript titled “Mitigating Co Metal Particle Agglomeration and Enhancing ORR Catalytic Activity through Nitrogen-Enriched Porous Carbon Derived from Biomass” presents a synthetic approach for the preparation of nitrogen-enriched soybean-derived porous carbon as an electrocatalyst for the ORR. This work contributes significantly to the field and offers promising and relevant findings within the scope of the Journal. However, there are several issues that need to be addressed before considering publication in Catalysts.

  1. Remove the phrase 'Type of the Paper' from the beginning of the manuscript.

Response: Thank you very much for your comment. 'Type of the Paper' from the beginning of the manuscript has been deleted.

  1. The authors should enhance the introduction by highlighting the significance of ZIG as precursors for synthesizing heteroatom-doped porous materials in this particular field.

Response: Thank you very much for your constructive comments. The significance of ZIFs as precursors for synthesizing heteroatom-doped porous materials in this particular field has been emphasized, please see “Introduction” section (page 2, line 86–92; page 3, line 102–108.).

  1. In my opinion, Figure 1 should be included under the Methodology section.

Response: Thanks a lot for the comments. According to the requirement of reviewer, the Figure 1 has been adjusted to the appropriate position in the revised version of the manuscript.

  1. Could you please provide the size of Co@Sp-C-a? Additionally, it would be valuable to discuss the conclusions that can be drawn from the XRD patterns, SEM, and TEM images.

Response: Thank you very much for your constructive comment. The particle size distribution of Co@Sp-C-a has been provided, please see Figure 2c. In addition, the descriptions of XRD, SEM, and TEM from “Characterization of Morphology and Structure” section have made appropriate modifications and supplements. Please refer to page 6, line 233-256 of the text for detailed modification.

  1. What was the rationale behind selecting a scan rate of 50 mV/s for analyzing the electrochemical properties, as opposed to 10 or 20 mV/s? Similarly, what motivated the choice of 0.1 M KOH electrolyte? These aspects warrant further clarification in the manuscript.

Response: Many thanks for your careful review and comments. More detailed explanation of “Electrochemical measurements” section has been supplemented (page 5, line 197–204.). In addition, 0.1 M KOH is a common electrolyte concentration for ORR testing to simulate alkaline environments. The mass transfer resistance of oxygen will increase if the concentration is too high.

  1. Could you please provide more information regarding the current density target of the process? Elaborating on this important aspect would enhance the understanding of the research findings.

Response: Thank you very much for your comments. In order to have a clearer understanding of this work, more detailed explanation of “Electrochemical measurements” section (page 5, line 197–219.) and “Electrochemical properties” section has been supplemented (page 9, line 325–352.).

  1. It is recommended to enhance the quality of the figures presented in the manuscript to improve their clarity and visual appeal.

Response: Thank you very much for your kind reminding. According to the requirement of reviewer, the resolution of the figures in the main text has been improved.

  1. The revised version of the manuscript should include a dedicated materials subsection to provide detailed information about the materials used in the study.

Response: Many thanks for your careful review and comments. More detailed explanation of “Materials and Methods” section has been supplemented (page 3-5, line 127–219.).

  1. Could you please provide an explanation for the observed improvement in results compared to Pt-based catalysts? It would be highly informative to understand the underlying phenomena driving this process, including aspects such as the Tafel slope, mass activity, and other relevant parameters.

Response: Thank you very much for your constructive comment. Tafel slope and electrochemical double-layer capacitance (Cdl) data of SP-C, SP-C-a, and Co@SP-C-a have been supplemented. Please refer to the revised version of the manuscript (Figure 4d, page 9, line 344–352.) and supporting information for details (Figure S9- S10).

  1. It is recommended to include additional references in the revised version of the manuscript to further support the presented findings.

Response: Thanks for the reviewer’s suggestion. The valuable references have been added and cited at appropriate place for the revised manuscript, please see “Introduction” section (References 21–24.), “Materials and Methods” section (References 30–31.) and “Electrochemical properties” section (References 44,46–47.).

  1. Could you please provide information about the composition of the electrocatalyst?

Response: Thanks a lot for the comments. From the XRD, HRTEM, and XPS data, it can be inferred that Co@SP-C-a catalyst contains metal cobalt wrapped in carbon and cobalt oxide due to partially oxidized by air.

Round 2

Reviewer 2 Report

The authors have thoroughly and accurately addressed all the questions.